# Novel Design and Finite Element Analysis of Diamond-like Porous Implants with Low Stiffness

**DOI:** 10.3390/ma14226918

**Published:** 2021-11-16

**Authors:** Jinyang Zhang, Xiao Zhang, Yang Chen, Wei Feng, Xianshuai Chen

**Affiliations:** 1Foshan Angels Biotechnology Co., Ltd., Foshan 528000, China; jy.zhang@gzjanus.com; 2Guangzhou Janus Biotechnology Co., Ltd., Guangzhou 511458, China; 3Guangdong Provincial People’s Hospital, Guangzhou 510080, China; Krystal-L26@163.com; 4Shenzhen Institute of Advanced Technology, Chinese Academy of Sciences, Shenzhen 518000, China; zx623@126.com; 5Foshan Hemera Biotechnology Co., Ltd., Foshan 528000, China; fourior@foxmail.com

**Keywords:** porous dental implants, elastic modulus, finite element analysis, implant-bone interface, stress distribution

## Abstract

The purpose of this study was to design porous implants with low stiffness and evaluate their biomechanical behavior. Thus, two types of porous implants were designed (Type I: a combined structure of diamond-like porous scaffold and traditional tapered thread. Type II: a cylindrical porous scaffold filled by arrayed basic diamond-like pore units). Three implant-supported prosthesis models were constructed from Type I, Type II and commercial implants (control group) and were evaluated by finite element analysis (FEA). The stress distribution pattern of the porous implants were assessed and compared with the control group. In addition, the stiffness of the cylindrical specimens simplified from three types of implants was calculated. The Type I implant exhibited better stress distribution than the Type II implant. The maximum stress between the cortical bone–Type I implant interface was 12.9 and 19.0% lower than the other two groups. The peak stress at the cancellous bone–Type I implant interface was also reduced by 16.8 and 38.7%. Compared with the solid cylinder, the stiffness of diamond-like pore cylinders simplified from the two porous implants geometry was reduced by 61.5 to 76.1%. This construction method of porous implant can effectively lower its stiffness and optimize the stress distribution at the implant–bone interface.

## 1. Introduction

The dental implant can act as a natural tooth to withstand continuous static and dynamic loads during oral restoration [1]. The osseointegration performance of dental implants plays an important role in the success rate of implant surgery [2]. Studies have shown that the design of implants affects their osseointegration [3]. In general, the design and manufacture of traditional dental implants are usually based on solid structures with regular cylindrical, tapered or threaded shapes [4].

Titanium is considered as a perfect metal material in dental prostheses due to its outstanding biocompatibility, high biomechanical strength and good corrosion resistance [5,6]. However, since the elastic modulus of titanium implants (110 GPa) is much higher than that of the surrounding bone tissue (1–20 GPa), stress shielding effects are likely to occur around the implant [7]. This will cause surrounding bone resorption, and can even lead to implant failure [8,9,10,11]. Thus, porous materials have been developed to solve the issues that occur with bulk materials, which exhibit the ability to reduce the Young’s modulus [12]. A properly designed porous implant can accelerate the osseointegration process [13]. Studies suggest that tapered screw implant can form a strong anchoring force at the implant-bone interface to obtain good initial stability, while a porous implant design can provide a large surface area for the attachment and fusion of bone tissue [14]. Normally, the bone-implant contact area can be increased by roughening the implant surface or creating a porous scaffold structure [15]. There is clinical evidence that titanium alloy implants after surface roughening and porous design modification have shown a high clinical effect [16]. Due to the good match in elastic modulus between the porous implant and the bone, it is beneficial to reduce the stress shielding at the implant-bone interface, and the implant can also achieve good biological fixation [17]. Furthermore, the porous structure provides more space and surface area for the adhesion and growth of osteocytes, thereby increasing the bone binding rate.

The finite element method (FEM) is commonly used to simulate and analyze the biomechanical behavior of implant models with complex structures [18]. Its analysis results can provide guidance for the implant design and manufacture, such as biomechanical properties and bone-implant interface interaction [19]. Therefore, a large number of numerical calculations and studies have been conducted to evaluate the biocompatibility and mechanical properties of dental implants, such as bone formation, stress distribution at implant-bone interface. Desai et al. [20] analyzed the stress and strain contours of implants with different designs by the FEM, thus obtaining the optimal design of the implant. Chou and Müftü [21] used a finite element model to investigate the bone formation and healing mode after the implant placement. In the research of Yamanishil et al. [22], the neck design of the implants under oblique load was evaluated by FEM. Recently, Huang et al. [23] used a 3D finite element model to predict the bone ingrowth and tissue differentiation around different type of hollow porous implants.

The above research provided a wealth of theoretical knowledge for the design of porous implants through a large number of experiments or numerical calculations. However, further study on this new type of bionic implant with low elastic modulus is still necessary and desirable, including advanced design, biomechanical behavior, and the mechanical biological reaction of the bone tissue around the implant, which will provide a basic study for the clinical application of such implants. In this work, two new types of porous implants with different geometric shape were designed (Type I: a combined structure of diamond-like porous scaffold and traditional tapered thread feature; Type II: a cylindrical porous scaffold filled by arrayed basic diamond-like pore units). On the premise of keeping the biomechanical properties certain, they were designed to reduce stiffness, improve bone-implant contact areas and optimize the stress distribution at the bone-implant interface. The implants were evaluated by static mechanics simulation and compression stress–strain analysis.

## 2. Materials and Methods

### 2.1. Modeling of New-Type Porous Implants

Two new types of porous implants were proposed and established by using computer-aided design (CAD) software (Solidworks2021, Dassault Systèmes, Waltham, MA, USA). The average diameter and length of the implants were 4.0 and 10 mm, respectively (Figure 1). Both porous implants were designed based on the diamond-like unit cell strut diameter (D_S_) and pore size (D_P_), as shown in Figure 1a. In particular, the pore size is the diameter (D_S_) of the largest sphere that can be accommodated in the pore area of the diamond-like unit. The diamond unit cell has been used in orthopedic implant and bone tissue engineering scaffold design due to its special structure, which comprises 16 struts of equal length, with an angle of 109.5° between every two struts. This structure was isotropic and better adapted to multidirectional loading even with high porosity [13]. Furthermore, the structure of diamond unit cells is very similar to the composition of cancellous bone, and this structural similarity may facilitate bone ingrowth. In this study, the Type I porous implant was constructed by combining the porous scaffolds structure with traditional tapered thread characteristics, arranged in a thread-pore-thread sequence from top to bottom. The diamond-like porous scaffold portion was located in the middle of the implant, and the bottom and upper end structure were similar to the conventional tapered thread design to obtain good self-tapping and primary stability (Figure 1b). The Type II implant was designed as a cylinder porous scaffold filled by arrayed basic diamond-like pore units (Figure 1c). The porous implants with diameters of 300–900 μm are expected in bone tissue engineering [24]. The strut diameter and pore size of the two types of porous implants were designed to be 0.5 and 0.6 mm. A solid implant (non-porous implant) was selected as a control group for comparative analysis with the two diamond-like porous implants (Figure 1d). The detailed parameters of the implant models were obtained using modeling information, as shown in Table 1.

Three implant restoration 3D models based on the principle of surgical implantation were created using CAD software (Figure 2). The models simulate the implantation of a porous implant in alveolar bone tissue. A straight abutment with a height of 8 mm was tightly connected to the implant through a 3° Morse taper and a single ceramic prosthesis was fixed on the abutment. For the bone tissue, a 1 mm-thick cortical bone layer was around the cancellous bone [25].

### 2.2. Three-Dimensional Finite Element Analysis

The entire implant restoration model was imported into FEA software (Ansys Workbench 19.0, Swanson Analysis Inc., Houston, PA, USA) and meshed with tetrahedral elements. Through convergence analysis, the mesh sizes for all components and bone tissue were finally optimized to 0.6 and 0.2 mm. Based on the above setting conditions, 78,613–79,638 elements and 125,255–135,584 nodes were generated in all finite element models.

To simulate the average occlusal force in a natural and oblique direction, a static load of 100 N was applied on the crown surface at 45° to the long axis of the implants [26,27]. In this simulation, the interface between the bone tissue and the implant was assumed to consist of perfectly bonded contacts, without any displacement or rotation [28]. The nodes of the bone in all directions were fixed and its degree of freedom was equal to zero. The materials and their properties, namely the Young’s modulus and Poisson ratio, used in the simulation were: Ti-6A1-4V for the implants and abutments, 110 GPa, 0.321; zirconia for the crown, 205 GPa, 0.19; and cortical bone and cancellous bone (14.0 GPa), respectively.

### 2.3. Compression Test Simulation

The compression test simulation was performed to obtain the stiffness of the Ti-6Al-4V material with porous structures. According to the ISO 13314 standard, the schematic diagram of the compression test of cylindrical specimen is given (Figure 3). A vertical compressive force (F) was applied on the compression surface of the specimen. The effective stiffness (E) can be calculated by the following formula:E = (F/A) × (h/∆h)(1)
where F is the loading force, A is the initial area of the compression surface, and the axial strain is calculated as ε = h/∆h.

Considering the pore size and length of the porous implants, a cylinder with a diameter of 5 mm and a height of 10 mm was selected as the basic model in this compression simulation. The compression models of the Type I implant, Type II implant and control group were simplified into an intermediate porous cylinder, a porous cylinder and a solid cylinder, respectively (Figure 4a–c). A load force of 1000–9000 N was applied on the upper surface of the cylinders. The maximum deformations during the compression simulation were recorded as the ∆h of each cylinder.

## 3. Results

The Von Mises stress cloud diagrams of Type I, Type II and the control group are shown in Figure 5. The essential condition for successful long-term implantation is to prevent mechanical failure of the implant. The zone of high stress within the inter-pore area can cause the implant to fail [29]. For the case of the Type I implant, the stress was evenly distributed in the porous layer, and high stress was observed at the node where the struts cross, with a value of 475.27 MPa (Figure 5a). The peak stress (629.66 MPa) for the Type II implant was concentrated in the top pore layer, and the value gradually decreased towards the bottom of the implant (Figure 5b). However, the peak stress (379.35 MPa) for control group was found around the implant neck (Figure 5c), which was about 20 and 39.8% lower than that of the Type I and Type II implant. It was obvious that the general patterns for stress distribution were quite different for the two designs.

The stress distribution on the implant-bone interface is described in Figure 6. Under oblique occlusal loading, the cancellous bone exhibits low stress amplitude for its lower elastic modulus. The peak stress located in the cortical bone for the Type I implant was about 13% lower than that of the Type II implant, but both of them are lower than the control group. The lowest principal stress level of cancellous bone was observed in the Type I implant with a value of 5.48 MPa, which was slightly lower than that of the Type II implant (6.59 MPa) and the control group (8.93 MPa). For cancellous bone, the peak stress for Type I was observed around the second thread of the implant. However, the maximum stress for Type II occurred near the implant apex. In the control group, the cancellous bone around the apical section of the implant showed the greatest stress distribution.

The maximum deformations of cylindrical specimens during the compression simulation under the load force of 3000 N are presented in Figure 7. The maximum deformations ∆h of intermediate porous cylinder, porous cylinder and solid cylinder are ∆h_1_ = 0.0636 mm, ∆h_2_ = 0.205 mm and ∆h_3_ = 0.0245 mm, respectively. The static stiffness of cylindrical specimens under different load forces was calculated and compared (Figure 8). It indicates that the calculated stiffness of intermediate porous cylinder and porous cylinder was much lower than that of solid cylinder. The calculated average stiffness of the intermediate porous cylinder and the porous cylinder were about 37.5 and 23.3 GPa, which were much lower than that of the solid cylinder (97.3 GPa).

## 4. Discussion

### 4.1. Diamond-Like Porous Implant Design

Titanium implants with a low elastic modulus are more feasible regarding biomechanical compatibility and can reduce the stress shielding effect [30]. However, an implant with properly designed porous structures can exhibit a strain and elastic modulus appropriate to the bone [31]. The extensive literature indicates that the diamond-type cellular structure has been used in bone tissue engineering due to its similarity to the trabecular bone [32,33,34]. This emergent type of porous implant may offer the potential to mitigate stress shielding, offer selective bioactivity, and adapt to changes in macroscale bone geometry when adapting to the complex load [35]. Based on these biomechanical advantages, this article further integrates the porous structure with the universal tapered thread structure to design a dental implant for oral restoration. Thus, two new types of porous implants with different geometry and porous structure design were constructed based on the diamond-like unit cell. The numerical simulation analysis was performed to clarify the effect of porous implant geometry on biomechanical properties and how they affected static stiffness, so as to guide the optimum design of porous implants.

### 4.2. Static Mechanical Performance

FEM is a recognized theoretical approach for understanding the biomechanical behavior of implant restorations [36]. In this study, static mechanical analysis was performed to calculate the stress distribution of the porous implants with different geometries. The stress distribution at the implant-bone interface was characterized to better understand the effect of different porous implant geometries on the surrounding bone. The FEA results show that the Type I implant exhibited better stress distribution than the Type II implant by effectively combining a tapered thread shape and a porous structure regularly. Although their maximum stress values were higher than the non-porous control implant, they did not exceed the yield strength under the given applied load (the yield stress of Ti-6Al-4V is about 970 MPa). Furthermore, the stress distribution patterns were quite different for the porous implants and non-porous implant. The stress concentrations of both the Type I implant and the Type II implant were located at the node where the struts cross in the porous layer. However, there was a high stress level at the non-porous implant internal wall in contact with the abutment collar, which may induce local tensile stress and may even cause fracturing [37]. Obviously, through the porous structure design of the non-porous implant, the maximum stress concentration is transferred from the implant-abutment contact area to the porous structure, which reduces the failure risk of the abutment collar to a certain extent. For studying the stress distribution of the surrounding bone, the stress at the implant-bone interface was characterized and quantized. The analysis results indicate that the maximum stress value between the cortical bone-Type I implant interface was 12.9 and 19.0% lower than that of the Type II implant and the non-porous control implant. Moreover, the lowest principal stress level between the cancellous bone-implant interfaces was found in the Type I implant, which was 16.8 and 38.7% lower than that of the Type II and control implant.

Although the maximum stress of the Type I implant is slightly higher than that of the control group, the stress distribution of the surrounding bone is optimized. The maximum stress of cortical bone and cancellous bone is reduced compared to the Type II and control implants. Fatigue damage is most likely to occur at the implant wall and collar (the contact area between implant and abutment) under external loads [37,38]. However, it is worth mentioning that after the porous structure design (Type I implant), the stress concentration of the implant is transferred from the upper inner wall to the porous structure area. From this point of view, the type I implant design may obtain a better resistance to load at the implant-abutment interface and show more satisfactory fatigue performance.

### 4.3. Compression Performance

Considering the irregularity of the implant geometry, the stiffness of the simplified cylindrical models of the three implants was calculated numerically. The compression simulation of cylindrical specimens showed that the cylinders with a special porous structure exhibit much lower stiffness when compared to the solid cylinder. Through the design of the porous structure based on diamond-like pore units, the stiffness of the solid cylinder is reduced by about 61.5 to 76.1%. However, a decrease in implant stiffness can improve the stress distribution pattern and enhance the conditions for bone remodeling [12]. Under the same external load, the elastic deformation of the cylinders with a porous structure were significantly smaller than that of the solid cylinder, which indicated that the dental implant with such a porous design may obtain a lower elastic modulus value. The above analysis may prove that the elastic modulus of conventional implants can be reduced by designing such a porous structure so that it is closer to that of bone tissue, thus reducing stress shielding and bone resorption [39]. As described in previous studies, the specific elastic modulus suitable for the bone characteristics can be obtained through the cell structure design of Ti-6Al-4V [40,41].

### 4.4. Limitations and Future Developments

The present study attempted to provide a theoretical basis for the design of new porous implants through the establishment of accurate mathematical models and finite element analysis. It needs to be clarified that the present study is only part of the overall research, and the application of this new porous implant and other characteristics are still being investigated. Based on this new type of construction technique for diamond-like porous structures, various porous dental implants with low stiffness can be obtained. However, considering the manufacturability of the new type of porous implant, further study is necessary on its manufacturing process, which can be conducted using 3D metal printing technology. The static and compressive performance numerical analysis of the three implant design models preliminarily confirmed that the Type I porous implant design was superior to the other two implant designs. In order to support this point of view, further experimental investigation of the mechanical behavior of the three implants is needed, such as compression tests and dynamic fatigue tests.

Taking into account the oral bone regeneration performance of the porous implant, the potential influence of such a material on stem cells during bone healing should be evaluated [42]. In addition, the insertion torque also needs to be assessed to confirm its primary stability, and how such implants can modify prosthetic rehabilitation during immediate loading procedures [43,44].

## 5. Conclusions

Benefiting from the obtained results presented above, the following conclusions were drawn:A novel design method of a porous dental implant (Type I porous implant) was proposed by combining the diamond-like pore structure with traditional tapered thread characteristics.Compared with the solid implant, the stiffness of the diamond-like porous implant was significantly reduced. Furthermore, the Type I porous implant exhibited better biomechanical behavior and stress distribution than the completely porous structure design. In summary, the Type I implant is superior to the other two implant designs.

## Figures and Tables

**Figure 1 materials-14-06918-f001:**
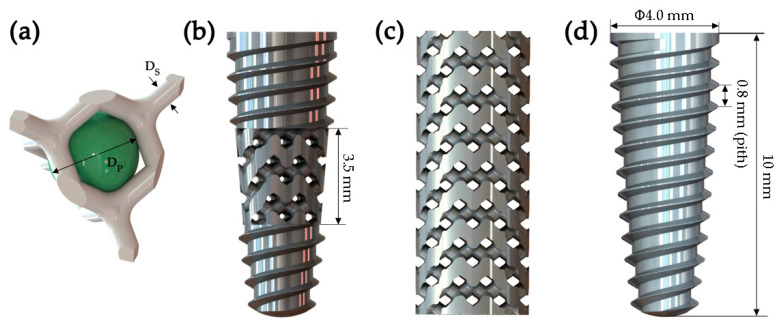
Geometrical models of dental implants: (**a**) single diamond-like unit cell (D_s_ and D_p_ are the strut diameter and pore size of the diamond-like unit cell, respectively), (**b**) Type I porous implant, (**c**) Type II porous implant, (**d**) control group.

**Figure 2 materials-14-06918-f002:**
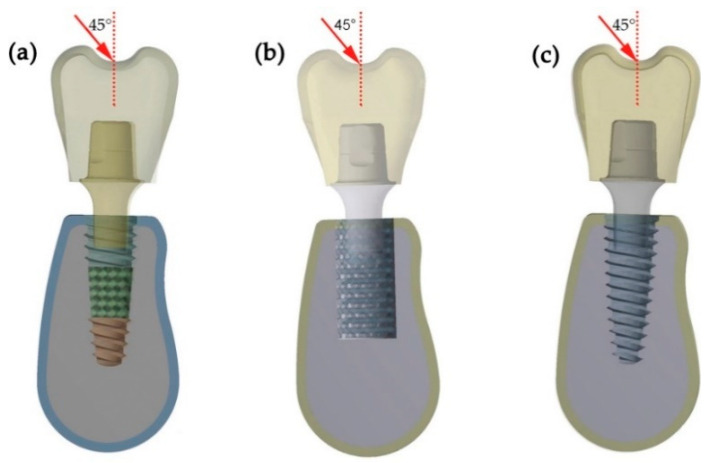
Dental implant restoration model in bone tissue: (**a**) Type I porous implant, (**b**) Type II porous implant, (**c**) control group.

**Figure 3 materials-14-06918-f003:**
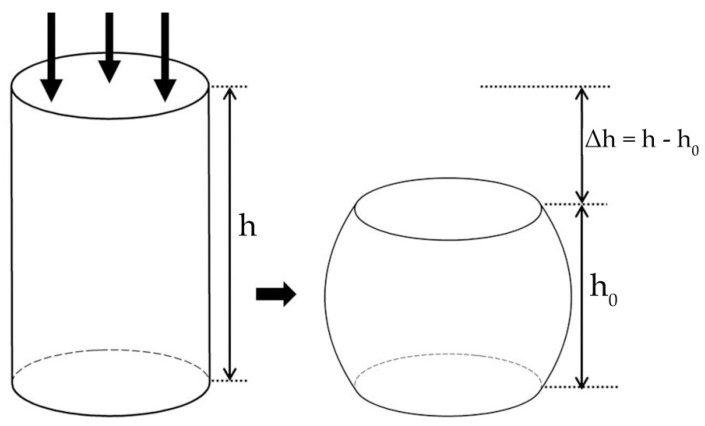
Schematic diagram of compression test for cylindrical specimen.

**Figure 4 materials-14-06918-f004:**
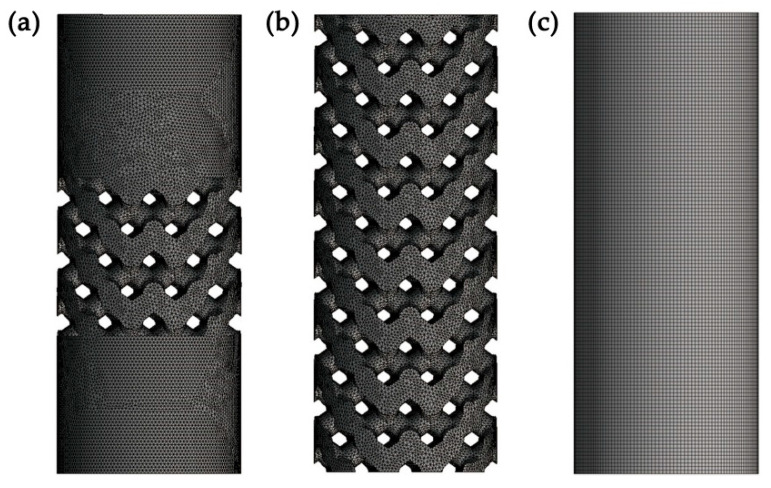
Finite element model of cylindrical specimens for compression simulation: (**a**) intermediate porous cylinder, (**b**) porous cylinder, (**c**) solid cylinder.

**Figure 5 materials-14-06918-f005:**
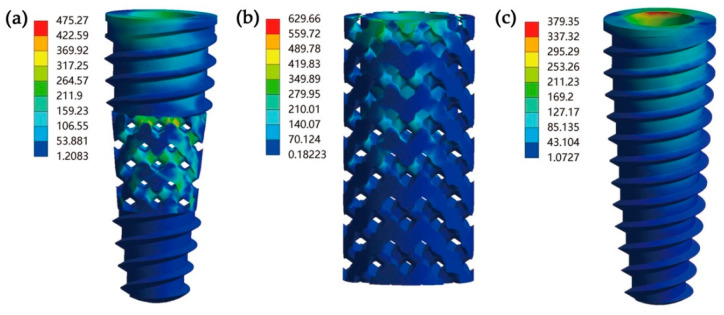
The Von Mises stress cloud diagrams of implants: (**a**) Type I implant, (**b**) Type II implant, (**c**) control group.

**Figure 6 materials-14-06918-f006:**
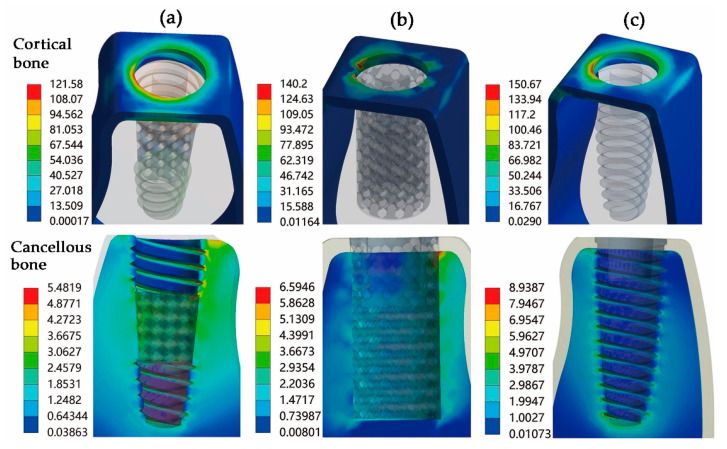
Stress distribution in cortical and cancellous bone: (**a**) Type I implant-bone interface, (**b**) Type II implant-bone interface, (**c**) control group.

**Figure 7 materials-14-06918-f007:**
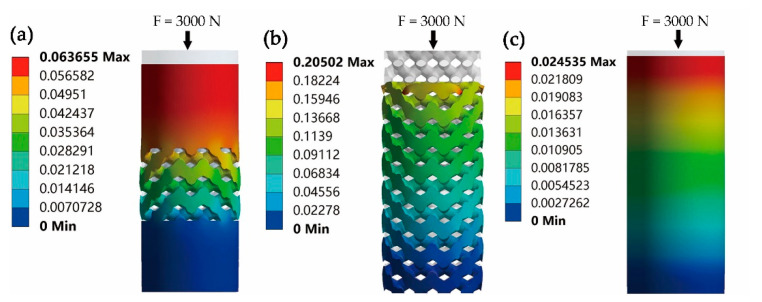
The maximum deformations (∆h/mm) of cylindrical specimens under the load force of 3000 N: (**a**) intermediate porous cylinder, (**b**) porous cylinder, (**c**) solid cylinder.

**Figure 8 materials-14-06918-f008:**
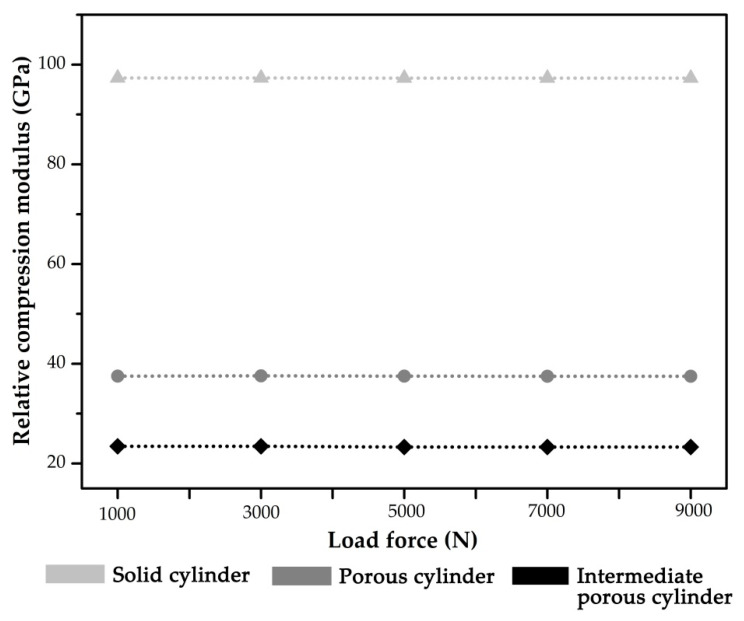
The stiffness of cylindrical specimens under different load forces.

**Table 1 materials-14-06918-t001:** The detailed geometric parameters of the implant models.

Model	Diameter (mm)	Length (mm)	Pore Size (mm)	Strut Diameter (mm)	Specific Surface Area (mm^3^)
Type Ⅰ	4.0	10	0.5	0.6	186.65
Type Ⅱ	4.0	10	0.5	0.6	429.18
Control group	4.0	10	-	-	142.07

## Data Availability

Data are contained within the article.

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
