# Peer review of "Novel Design and Finite Element Analysis of Diamond-like Porous Implants with Low Stiffness"

_materials, 2021, doi:10.3390/ma14226918_

Round 1

Reviewer 1 Report

The article is interesting but some issue should be clarified

Line 44 is an author affirmation or should provide a citation

Line 69-71 should be reformatted because is not clear.

line 84 please define a certain sequence

line 90 what is a strut diameter?

The discussion should be expanded. 

in particular authors could compare the findings and method with 

 Tribst JPM, dal Piva AMO, Giudice RL, Borges ALS, Bottino MA, Epifania E, et al. The influence of custom-milled framework design for an implant-supported full-arch fixed dental prosthesis: 3D-FEA sudy. Int J Environ Res Public Health 2020;17(11):1-12.

Lo Giudice R, Puleio F, Rizzo D, Alibrandi A, Lo Giudice G, Centofanti A, et al. Comparative investigation of cutting devices on bone blocks: An SEM morphological analysis. Appl Sci 2019;9(2).

Authors should better describe the bone model used

The discussion is not clear please reformat it to improve readability

A.A. think that model 2 could be ever produced or is just a concept to study the forces transmission? This could be discussed in text

The references should be formatted according to journal guidelines please review it for errors.

Author Response

Thank you for your comments concerning our manuscript. Those comments are all valuable and very helpful for revising and improving our paper, as well as the important guiding significance to our researches. We have studied comments carefully and have made correction which we hope meet with approval.

Reviewer #1:

  1. Response to comment: Line 44 is an author affirmation or should provide a citation

Response: Thank you for your comments. Line 44 has provided a citation.

  1. Response to comment: Line 69-71 should be reformatted because is not clear.

Response: Thank you for your comments. Line 69-71 has been reformatted.

  1. Response to comment: Line 84 please define a certain sequence

Response: Thank you for your comments. Line 84 has been modified.

  1. Response to comment: line 90 what is a strut diameter?

Response: Thank you for your comments. We have given a detailed explanation of strut diameter and pore size, and attached a schematic diagram of diamond-like unit cell, as shown in figure 1a.

  1. Response to comment: The discussion should be expanded. In particular authors could compare the findings and method with Tribst JPM, dal Piva AMO, Giudice RL, Borges ALS, Bottino MA, Epifania E, et al. The influence of custom-milled framework design for an implant-supported full-arch fixed dental prosthesis: 3D-FEA sudy. Int J Environ Res Public Health 2020;17(11):1-12. Lo Giudice R, Puleio F, Rizzo D, Alibrandi A, Lo Giudice G, Centofanti A, et al. Comparative investigation of cutting devices on bone blocks: An SEM morphological analysis. Appl Sci 2019;9(2).

Response: Thank you for your comments. The discussion has been expanded.

  1. Response to comment: The discussion is not clear please reformat it to improve readability

Response: Thank you for your comments. The discussion has been reformatted.

  1. Response to comment: A.A. think that model 2 could be ever produced or is just a concept to study the forces transmission? This could be discussed in text.

Response: Thank you for your comments. We have added 4.3 Limitations and Future Developments in the discussion section, which discussed the manufacturability of the porous implant and its manufacturing method.

  1. Response to comment: The references should be formatted according to journal guidelines please review it for errors.

Response: Thank you for your comments. After modification, the references have been formatted according to journal guidelines.

Reviewer 2 Report

The authors proposed a new dental implant design to reduce the elastic modulus and suppress stress shielding, and performed mechanical analysis by FEA. Unfortunately, I could not find any novelty in this content. First of all, what is the concept of the design? If the concept is only to reduce the elastic modulus by creating a porous material, then any porous structure can be accepted. For publication, the manuscript need an original concept that is specific to this research. Furthermore, if dental implant bodies that are subjected to severe loading are made by porous materials, can they withstand fatigue? Are the authors only focusing on reducing the modulus and ignoring other important aspects?

Author Response

Thank you for your comments concerning our manuscript. Those comments are all valuable and very helpful for revising and improving our paper, as well as the important guiding significance to our researches. We have studied comments carefully and have made correction which we hope meet with approval.

Reviewer #2:

Response to comment: First of all, what is the concept of the design? If the concept is only to reduce the elastic modulus by creating a porous material, then any porous structure can be accepted. For publication, the manuscript needs an original concept that is specific to this research. Furthermore, if dental implant bodies that are subjected to severe loading are made by porous materials, can they withstand fatigue? Are the authors only focusing on reducing the modulus and ignoring other important aspects?

Response: Thank you for your comments. The design concept of this porous implant is to integrate the diamond-like unit cell structure with the traditional tapered thread structure, so as to obtain a new porous implant with good self-tapping, primary stability and novel bionic porous structure. We have expanded section 2.1 (Modeling of New-type Porous Implants) and introduced the design idea of the porous implant in detail. According to the current statics simulation analysis, the maximum stress of the Type I porous implant under the same load is not much different from that of the solid implant. In the next work, we will study the manufacturing process of porous implants and their dynamic fatigue and other biomechanical properties. These considerations are discussed and explained in section 4.3 Limitations and Future Developments.

Reviewer 3 Report

The paper submitted for review presents two new design solutions for dental implants with a porous structure. While the concept proposed by the authors is very interesting, the rest of the work is not free from numerous ambiguities and errors. In particular, the authors' basic assumption that the porous part of the structure can be treated in terms of mechanical parameters as the fully dense Ti – 6Al – 4V alloy itself is incorrect. This in turn leads to incorrect calculations and conclusions. It is a pity that the authors did not use, referring to the work [32], that we have the following properties: Es, ρs and σs are the elastic modulus, density and yield strength of fully dense solid materials, and E, ρ and σ are the apparent modulus, density and yield strength of open-cell cellular structures [32]. Below are the detailed comments made by the work:

  1. Modify the title - remove the repetition.
  2. A certain abuse in this case is the term diamond-like structures, as such structures are usually considered at the nano level, for example in the case of carbon coatings (Crystalline structure and properties of diamond-like materials - DOI 10.17586 / 2220-8054-2017 -8-1-127-136).
  3. In the line 17 we found "control group" - what does it mean? 4. The authors refer to the publication from 2003. A lot has changed since then, which you can read about in: Porous Titanium Implants: A Review - DOI: 10.1002 / adem.201700648.
  4. The authors refer to the publication from 2003. A lot has changed since then, which you can read about in: Porous Titanium Implants: A Review - DOI: 10.1002 / adem.201700648.
  5. The authors assumed a mesh size at the level of 0.2-0.6 mm, i.e. close to the pore size, which makes it difficult to assess stresses and strains in the FE simulation near the pores.
  6. The authors assumed the force loading the implant at the level of 100 N. This value is significantly underestimated (what is the source of this value?).
  7. Relative elastic modulus is determined for the load of the entire structure, which is not correct as it is not homogeneous in terms of material. Hence, the fact emphasized by the authors that the new constructions have a lower value of Young's modulus is not correct. In this case, it would be more correct to use the term stiffness, which is defined for the entire structure of the implant.
  8. The phrase "stress between" (line 162) is not correct, because we do not assess the stress between the implant and bone tissue, but in the tissue or in the implant.
  9. Typo in Figure 6a - it should be "cortical".

In conclusion: as it stands, an article is not eligible for publication, unless the authors make major changes.

Author Response

Thank you for your comments concerning our manuscript. Those comments are all valuable and very helpful for revising and improving our paper, as well as the important guiding significance to our researches. We have studied comments carefully and have made correction which we hope meet with approval.

Reviewer #3:

  1. Response to comment: Modify the title - remove the repetition.

Response: Thank you for your comments. The repetition in title has been removed.

  1. Response to comment: A certain abuse in this case is the term diamond-like structures, as such structures are usually considered at the nano level, for example in the case of carbon coatings (Crystalline structure and properties of diamond-like materials - DOI 10.17586 / 2220-8054-2017 -8-1-127-136).

Response: Thank you for your comments. In recent years, the diamond-like structure has been used in orthopedic implants and bone tissue engineering scaffolds designing. For example in the case of Ti6Al4V scaffolds/ implants. (Bone bonding strength of diamond-structured porous titanium-alloy implants manufactured using the electron beam-melting technique- Doi.org/10.1016/j.msec.2015.11.025/ The biomimetic design and 3D printing of customized mechanical properties porous Ti6Al4V scaffold for load-bearing bone reconstruction- Doi:10.1016/j.matdes.2018.04.065)

  1. Response to comment: In the line 17 we found "control group" - what does it mean?

Response: Thank you for your comments. The solid implants (non-porous implant) are used as a control group for to compare and confirm whether the porous design has changed in terms of stress distribution and compression performance.

  1. Response to comment: The authors refer to the publication from 2003. A lot has changed since then, which you can read about in: Porous Titanium Implants: A Review - DOI: 10.1002 / adem.201700648.

Response: Thank you for your comments. Lines 35-38 are not rigorous and have been modified to “In general, the design and manufacture of traditional dental implants are usually based on solid structures with regular cylindrical, tapered or threaded shapes”.

  1. Response to comment: The authors assumed a mesh size at the level of 0.2-0.6 mm, i.e. close to the pore size, which makes it difficult to assess stresses and strains in the FE simulation near the pores.

Response: Thank you for your comments. The pore size of two types of porous implants was 0.6 mm. In this simulation analysis, the mesh size of porous structure is set to 0.2 mm. The mesh size of other components is set to 0.6 mm, such as ceramic prosthesis, etc. The results are convergent under this setting and can effectively evaluate the stress around the porous.

  1. The authors assumed the force loading the implant at the level of 100 N. This value is significantly underestimated (what is the source of this value?).

Response: Thank you for your comments. Load in the mouth is complex and are rarely transmitted vertically, and its normal bite force range is 30-300 N. A 30°/45° oblique load of 100N is often used in the simulation analysis of implants. (Tribst JPM, dal Piva AMO, Giudice RL, Borges ALS, Bottino MA, Epifania E, et al. The influence of custom-milled framework design for an implant-supported full-arch fixed dental prosthesis: 3D-FEA sudy. Int J Environ Res Public Health 2020;17(11):1-12. / Bordin D ,  Bergamo E ,  Fardin V P , et al. Fracture strength and probability of survival of narrow and extra-narrow dental implants after fatigue testing: In vitro and in silico analysis. Journal of the Mechanical Behavior of Biomedical Materials, 2017, 71:244-249.)

  1. Response to comment: Relative elastic modulus is determined for the load of the entire structure, which is not correct as it is not homogeneous in terms of material. Hence, the fact emphasized by the authors that the new constructions have a lower value of Young's modulus is not correct. In this case, it would be more correct to use the term stiffness, which is defined for the entire structure of the implant.

Response: Thank you for your comments. The expression in the text is not rigorous enough. We have corrected the relative elastic modulus to relative compression modulus.

      8. Response to comment: The phrase "stress between" (line 162) is not                      correct, because we do not assess the stress between the implant and                  bone tissue, but in the tissue or in the implant.

Response: Thank you for your comments. The expression in the text is not rigorous enough. The phrase "stress between has been modified to the stress in the cortical bone/cancellous bone.

       9. Response to comment: Typo in Figure 6a - it should be "cortical".

Response: Thank you for your comments. Spelling errors, figure 6a has been corrected.

Round 2

Reviewer 1 Report

If the paper have been compare to 

 Tribst JPM, dal Piva AMO, Giudice RL, Borges ALS, Bottino MA, Epifania E, et al. The influence of custom-milled framework design for an implant-supported full-arch fixed dental prosthesis: 3D-FEA sudy. Int J Environ Res Public Health 2020;17(11):1-12.

Lo Giudice R, Puleio F, Rizzo D, Alibrandi A, Lo Giudice G, Centofanti A, et al. Comparative investigation of cutting devices on bone blocks: An SEM morphological analysis. Appl Sci 2019;9(2).

The paper should be cited in the reference section. 

Author Response

Response to comment: Tribst JPM, dal Piva AMO, Giudice RL, Borges ALS, Bottino MA, Epifania E, et al. The influence of custom-milled framework design for an implant-supported full-arch fixed dental prosthesis: 3D-FEA sudy. Int J Environ Res Public Health 2020;17(11):1-12.

Lo Giudice R, Puleio F, Rizzo D, Alibrandi A, Lo Giudice G, Centofanti A, et al. Comparative investigation of cutting devices on bone blocks: An SEM morphological analysis. Appl Sci 2019;9(2).

The paper should be cited in the reference section. 

Response: Thank you for your comments. In section 2.2, the magnitude and loading direction of the occlusal force in the finite element analysis are compared with Tribst JPM, dal Piva AMO, Giudice RL, Borges ALS, Bottino MA, Epifania E, et al. The influence of custom-milled framework design for an implant-supported full-arch fixed dental prosthesis: 3D-FEA sudy. Int J Environ Res Public Health 2020;17(11):1-12, which has been cited in the reference section [27].

Reviewer 2 Report

Thank the authors for your response. But unfortunately, it has not led to a change in my opinion. The authors cited several new references on diamond-shaped porous structures. This means that the originality of this structure itself is low. Furthermore, in these literatures, the structure has been validated by mechanical and biological tests. FEA simulations should be used to screen the structure, and subsequent validation is mandatory.

Author Response

Response to comment: Thank the authors for your response. But unfortunately, it has not led to a change in my opinion. The authors cited several new references on diamond-shaped porous structures. This means that the originality of this structure itself is low. Furthermore, in these literatures, the structure has been validated by mechanical and biological tests. FEA simulations should be used to screen the structure, and subsequent validation is mandatory.

Response: Thank you for your comments. Your opinions are of great guiding significance to our future research work. Thank you again!

Reviewer 3 Report

Dear Authors,

I am satisfied with the authors' answers. I believe that in its present form article can be published.

Regards

Author Response

Response to comment: I am satisfied with the authors' answers. I believe that in its present form article can be published.

Response: Thank you for your comments. Thank you for your affirmation of the revised manuscript.